# Orientation processing by synaptic integration across first-order tactile neurons

**Etay Hay** [1,2,3]*, **J. Andrew Pruszynski** [1,2,3]

**1** Department of Physiology and Pharmacology, Western University, London, Canada, **2** Brain and Mind Institute, Western University, London, Canada, **3** Robarts Research Institute, Western University, London, Canada

* etay.hay@camh.ca

## Abstract

Our ability to manipulate objects relies on tactile inputs from first-order tactile neurons that innervate the glabrous skin of the hand. The distal axon of these neurons branches in the skin and innervates many mechanoreceptors, yielding spatially-complex receptive fields. Here we show that synaptic integration across the complex signals from the first-order neuronal population could underlie human ability to accurately (< 3˚) and rapidly process the orientation of edges moving across the fingertip. We first derive spiking models of human first-order tactile neurons that fit and predict responses to moving edges with high accuracy. We then use the model neurons in simulating the peripheral neuronal population that innervates a fingertip. We train classifiers performing synaptic integration across the neuronal population activity, and show that synaptic integration across first-order neurons can process edge orientations with high acuity and speed. In particular, our models suggest that integration of fast-decaying (AMPA-like) synaptic inputs within short timescales is critical for discriminating fine orientations, whereas integration of slow-decaying (NMDA-like) synaptic inputs supports discrimination of coarser orientations and maintains robustness over longer timescales. Taken together, our results provide new insight into the computations occurring in the earliest stages of the human tactile processing pathway and how they may be critical for supporting hand function.

## Author summary

Our ability to manipulate objects relies on tactile inputs signaled by first-order neurons that innervate mechanoreceptors in the skin of the hand and have spatially-complex receptive fields. Here we simulated populations of model human first-order neurons to show how synaptic integration across the rich inputs they provide can rapidly and accurately process the orientation of edges moving across the fingertip. We examined different types of synaptic integration to provide mechanistic insight into how higher-order neurons extract meaningful tactile information from the complex responses of the peripheral neuronal population. We thus show that synaptic integration across first-order neurons could underlie human ability to process edge orientations with high acuity and speed.

**Data Availability Statement:** All relevant data are within the manuscript and its Supporting Information files.

**Funding:** This work was supported by the Canadian Institutes of Health Research (Foundation

Grant to JAP: 353197) and the Government of
Ontario (Early Researcher Award to JAP). EH
received a postdoctoral fellowship from the Brain
and Mind Institute at Western University. JAP
received a salary award from the Canada Research
Chairs Program. The funders had no role in study
design, data collection and analysis, decision to
publish, or preparation of the manuscript.

**Competing interests:** The authors have declared
that no competing interests exist.

## Introduction

Tactile input from the hands is important for many behaviors, ranging from daily motor tasks
like buttoning a shirt to complex skills like knitting a sock [1]. This tactile information is con-
veyed by four types of first-order tactile neurons that innervate distinct mechanoreceptive end
organs in the glabrous skin of the hand [2]. Of particular note are fast-adapting type 1 neurons
(FA-1) innervating Meissner corpuscles and slow-adapting type 1 neurons (SA-1) innervating
Merkel discs, which are important for extracting fine spatial details of touched objects [3]. A
fundamental feature of FA-1 and SA-1 neurons is that their axon branches in the skin and
innervates many mechanoreceptors, ~30 on average [4], and thus these neurons have spa-
tially-complex receptive fields with many highly-sensitive zones [3,5–7]. We have recently pro-
posed that this innervation pattern may constitute a peripheral neural mechanism for
extracting geometric features of touched objects [6].

 Here we examine how the geometric features of touched objects are encoded by the popula-
tion of first-order tactile neurons and the synaptic readout of their activity. Our approach adds
to previous studies [8–13] in several ways. First, our models are based on first-order tactile
neurons that have spatially complex receptive fields, and involve spiking and other physiologi-
cal details that were absent in previous simple models of tactile receptive field responsivity [6].
Second, our models are constrained by neural recordings in humans, whereas previous models
used data from macaque monkeys [9,10]. Third, previous models of the peripheral neuronal
population innervating the fingertips [8,11,13,14] examined encoding of stimuli using abstract
features of its response, such as mean firing rate, spike count or first-spike latency. Other mod-
els examined the evolution of synaptic weights between first- and second-order neurons over
stimulus presentations but did not address the encoding of the stimuli [12]. In contrast, our
models examine the encoding of tactile stimuli using synaptic integration, and thus provide
mechanistic insight into how higher-order neurons could extract meaningful tactile informa-
tion from the complex responses of the peripheral neuronal population.

 We derive spiking models of first-order tactile neurons that fit and predict empirical data
with good accuracy. We then simulate the first-order neuronal population innervating a fin-
gertip, and investigate the computational capabilities afforded by synaptic integration of the
population activity. We show that synaptic integration across first-order neurons can account
for the human ability to discriminate tactile edge orientations with high acuity ($< 3°$) and
speed [15]. Our results suggest that discriminating edge orientation in this manner relies on a
small number of key inputs from first-order neurons. Our results also suggest that integrating
fast-decaying (AMPA) synaptic input over a short time scale (i.e. coincidence detection) is crit-
ical for robustly discriminating with high acuity whereas integrating slow-decaying (NMDA)
synaptic input serves to support the discrimination of coarser orientations and maintain
robustness over longer time scales (i.e. temporal summation).

## Results

### Data-driven models of first-order tactile neurons

We generated spiking models of human FA-1 neurons (n = 15 neurons) constrained by their
response to oriented edges moving across the fingertip. Each model neuron innervated a set of
mechanoreceptors, with an axonal branch and spike initiation zone at each mechanoreceptor
(Fig 1A). The input from each model mechanoreceptor depended on stimulus amplitude and
distance (see Methods). Spikes were generated at each axon initiation zone following a linear
relationship between mechanoreceptor input and axonal spike rate, along with a spike rate sat-
uration (see Methods). The model neuron output followed a "reset" scheme, whereby spikes

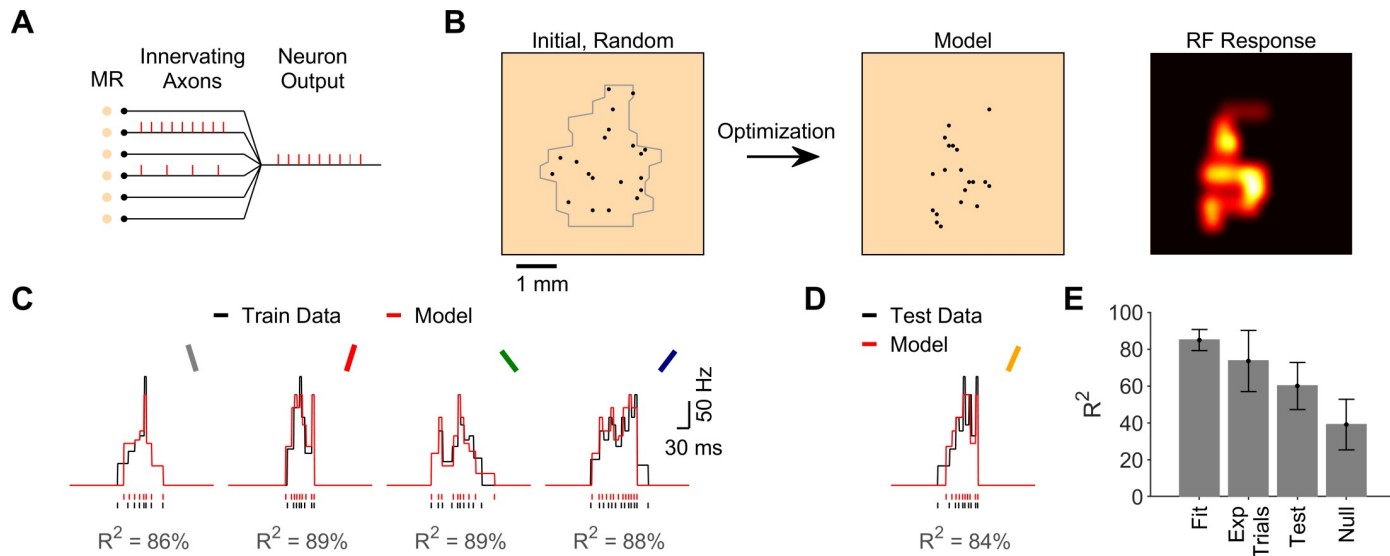

**Fig 1. Data-driven models of first-order tactile neurons. A.** The model neuron innervated a set of mechanoreceptors (MRs), each with its own axonal branch and spiking zone. The model neuron output followed a "reset" scheme, whereby spikes triggered at one axonal spiking zone reset initiation at other spiking zones. **B.** The locations of innervated mechanoreceptors (black) were derived by the model optimization algorithm. At the start of the optimization, locations were random within the area derived from the recorded neuron's response to a moving dot stimulus (gray, see Methods). The resulting model innervation pattern and receptive field (RF) response map are shown for the example FA-1 neuron whose response is illustrated in parts C and D. **C.** Fitness ($R^2$) of a model for an example FA-1 neuron. Observed (black) and model (red) spike trains and rate curves in response to edges oriented -22.5, 22.5, -45, and 45˚. **D.** Observed and model response to an edge oriented 30˚, which served to test the models. **E.** Model fitness, $R^2$ between different trials of the recorded neuron's response to the test orientation (30˚), model prediction accuracy for the test orientation (30˚), and the prediction accuracy of null models (shuffled, see Methods) across all neurons (n = 15). Prediction accuracy of the models was significantly higher than that of null models ($p < 10^{-4}$, paired-sample t-test). Error bars depict standard deviation.

initiated at one initiation zone reset the other spike initiation zones (Fig 1A). The free parameters of the model neuron were the locations of the innervated mechanoreceptors (Fig 1B).

We constrained the model for each recorded neuron using empirical spike responses to edges at four different orientations relative to their movement direction (±22.5, ±45˚). We used a genetic algorithm to derive the mechanoreceptor locations that yielded the best fit of model neuron response to the experimental response (see Methods). We optimized models using different number of mechanoreceptors (10–40) and cross-validated the models using a different edge orientation (0˚) that was not used during the model fitting. We selected the model that best fitted the training data and had the best cross-validation accuracy on the cross-validation data, with the fewest number of mechanoreceptors (see Methods). We then tested the model prediction on a different edge-orientation (30˚) that was not used in the model optimization. The model neurons fit the data with high accuracy ($R^2 = 85 \pm 6\%$, mean ± SD, n = 15 neurons, Fig 1C and 1E), had a good cross-validation accuracy ($R^2 = 65 \pm 17\%$), and predicted the test data similarly well ($R^2 = 60 \pm 13\%$, Fig 1D and 1E). The $R^2$ between the different trials of each recorded neuron's response to the test orientation was 74 ± 17% (Fig 1E, statistics calculated over 7 trials), and thus the model prediction accuracy was ~80% of the experimental response reliability.

The model prediction accuracy was significantly better than the accuracy of null models in which the mechanoreceptor locations were shuffled across the recorded neuron's receptive field boundary ($R^2 = 39 \pm 14\%$, n = 15, $p < 10^{-4}$, paired-sample t-test, Fig 1E). Prediction accuracy was also significantly better than when simply using the experimental response of the nearest edge (22.5˚) to predict response to the test edge (30˚, $R^2 = 47 \pm 19\%$, n = 15, $p < 10^{-2}$, paired-sample t-test). Models for the different neurons required 20 ± 5 mechanoreceptors to reproduce the recorded responses. Our model neurons using spiking and resetting, in

agreement with known physiology (see Methods), were thus able to fit and predict single trial spike response data with high accuracy, and therefore suggest that the diverse response of the neurons does not have to rely on analogue summation used in previous simpler models of human tactile receptive field responsivity [6].

## Encoding edge orientation via synaptic integration of first-order neuronal population activity

We next determined whether synaptic integration across the activity of FA-1 neuronal population can be used to discriminate edge orientations accurately. We simulated a model population of 330 FA-1 neurons innervating a 12x12 mm patch of skin, with model neuronal population generated as randomly-rotated variations of the 15 model neurons (see Methods). We simulated the population response to oriented edges moving over the skin patch, during a task of edge-orientation discrimination (Fig 2A). The task was to discriminate between two edges oriented at ±θ (where θ = 1, 3, 5, 10, or 20˚). The response of the first-order population was then convolved with a postsynaptic potential (PSP) waveform and fed into a classifier with two units (tuned to θ and -θ) that performed synaptic integration across the neuronal population (see Methods).

We inspected the discrimination accuracy of the classifiers using synaptic integration of PSPs of different decay time constants (see Methods)–either a short time constant corresponding to AMPA synapses ($\tau_{decay}$ = 3 ms), a long time constant corresponding to NMDA synapses ($\tau_{decay}$ = 65 ms), or a combination of both (Fig 2B). We also allowed negatively-weighted synapses with short and long time constants in the classifiers, which would correspond to feedforward $GABA_A$ and $GABA_B$ inhibition, respectively. We inspected the discrimination accuracy under different levels of additive stimulus noise (0–10%, see Methods). We trained the classifiers using a genetic algorithm to find the weights of the synaptic inputs from first-order neurons that yielded the best discrimination of edge orientations.

In the noiseless case, we found that synaptic integration of the activity in the first-order population could be used to discriminate different edge orientations perfectly with high acuity (to within ±1˚), for either of the PSP time constants (Fig 2C and 2D). When adding noise to the stimulus, the discrimination accuracy remained high (80–90% success), although discrimination became challenging for finer orientations. Discrimination accuracy using synaptic integration with NMDA synapses was more robust to noise compared to using AMPA synapses (improving test performance by 10–20%, Fig 2E). Synaptic integration using a combination of AMPA and NMDA PSPs did not significantly improve the robustness to noise.

We next varied the stimulus presentation time window (5, 10, 20, 50 ms, and unlimited, see Methods) and examined its effect on discrimination accuracy. In the noiseless case, the discrimination accuracy remained perfect regardless of the synapse type or time window size. When noise was applied, synaptic integration over shorter time windows increased the robustness of discrimination of classifiers that used fast-decaying (AMPA) synapses, particularly for the fine angles (1 and 3˚, Fig 3A). In contrast, integration time window did not significantly affect discrimination in classifiers that integrated slow-decaying (NMDA) synapses (Fig 3B). Overall, performance of classifiers that integrated population activity using AMPA synapses increased significantly in the case of short time window (5 ms, Fig 3D) compared to long time window (> 100ms, Fig 2C), approaching the robustness to noise of NMDA synapses over all orientations and improving performance for fine orientations (Fig 3E and 3F).

To inspect the contribution of first-order neurons to discrimination of different edges, we examined their synaptic weights and spatial layout in the different classifiers. Synaptic weights from most first-order neurons were close to 0 on average, and only a small subset of model

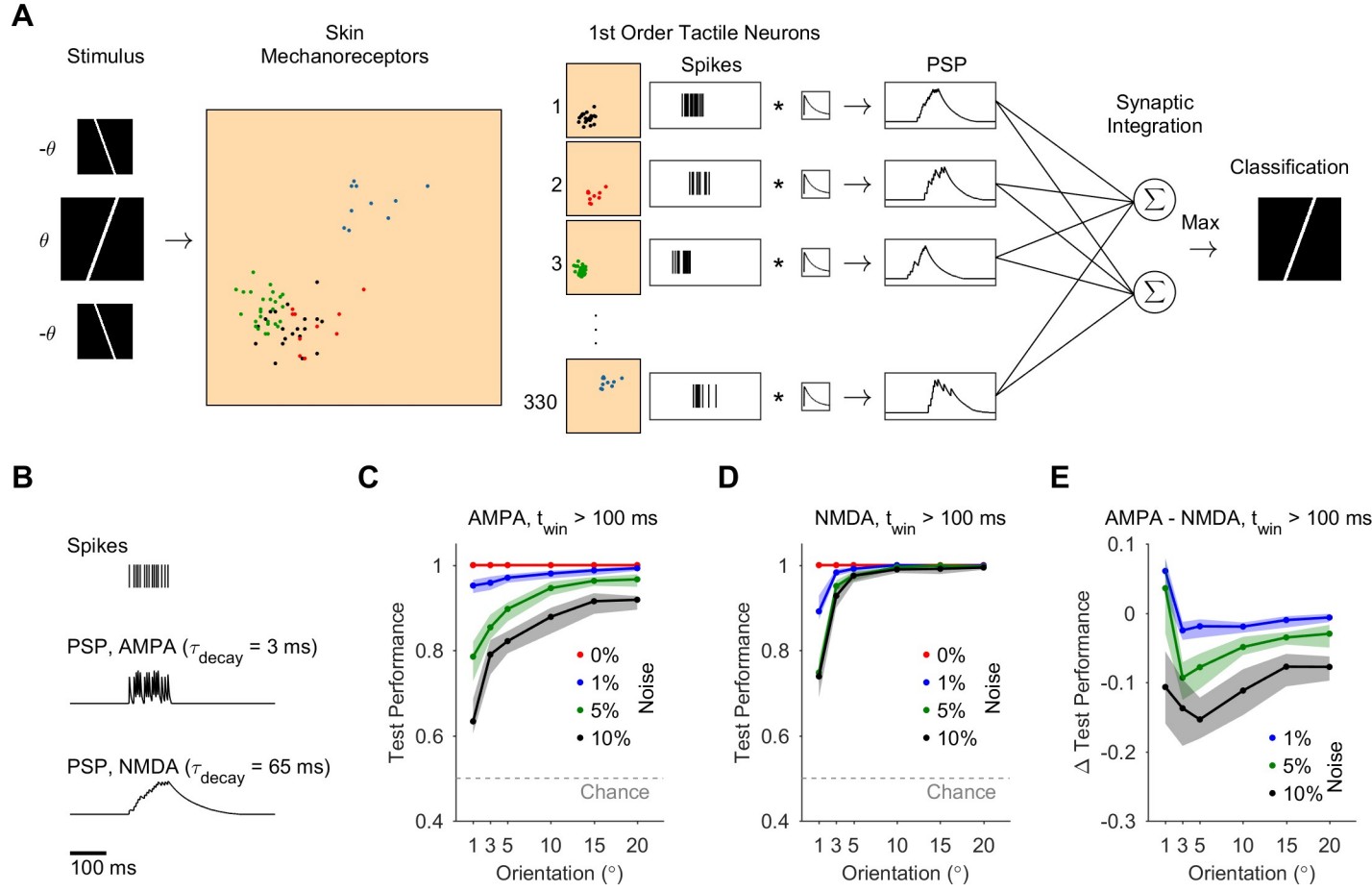

**Fig 2. Discrimination of edge orientation using synaptic integration of first-order neuronal population activity. A.** Simulated model population of FA-1 neurons innervating mechanoreceptors in a 12x12 mm patch of skin, during an edge-orientation discrimination task (-θ vs θ, θ = 1–20˚). Four example model first-order neurons are shown as color-coded sets of innervated mechanoreceptors, with their corresponding spike trains to the right. The neuronal population activity was fed into edge-orientation classifiers that performed synaptic integration of the neuronal population activity. The classifiers comprised of two units (tuned to -θ or θ) that performed a weighted sum of the synaptic inputs. Edge-orientation classification was determined according to the unit with maximal PSP value. **B.** PSPs were produced by convolving model first-order neuronal spike trains with a PSP waveform using either a short time constant (3 ms), corresponding to AMPA synapses, or a long time constant (65 ms), corresponding to NMDA synapses. **C, D.** Discrimination of test performance using synaptic integration with AMPA (C) or NMDA (D) synapses. Each curve corresponds to a different level of stimulus noise (between 0 and 10%), and shows mean and 95% confidence intervals. Dashed gray line shows chance level (0.5). **E.** The difference in test performance between classifier units using AMPA and NMDA synapses, for noise level 1, 5, 10%.

first-order neurons consistently provided large contribution to classification (Fig 4A, based on 95% confidence intervals estimated by bootstrapping synaptic weights from 20 randomized trained classifiers, corrected for multiple comparisons). The number of key contributors was 11 ± 3 first-order neurons on average in classifiers using fast-decaying (AMPA) synapses, and similar (8 ± 2) in classifiers using slow-decaying (NMDA) synapses (Fig 5A). We found no significant correlation between the edge orientation and the number of key synapses ($r = 0.27$, $p = 0.6$ for classifiers using AMPA synapses; $r = 0.44$, $p = 0.38$ for classifiers using NMDA synapses). The weights of the key excitatory synapses were anticorrelated between classifier units tuned to opposite orientations ($r = -0.81 ± 0.09$ across classifiers using AMPA synapses; $r = -0.79 ± 0.03$ across classifiers using NMDA synapses). An example of the key synaptic weights in classifiers integrating AMPA synaptic inputs for discriminating -20 and 20˚ is shown in Fig 4B ($r = -0.96$, $p < 10^{-4}$). A similar example for classifiers using NMDA synapses is shown in Fig 5B ($r = -0.88$, $p < 10^{-2}$). This was seen also in the receptive field map of the classifier units,

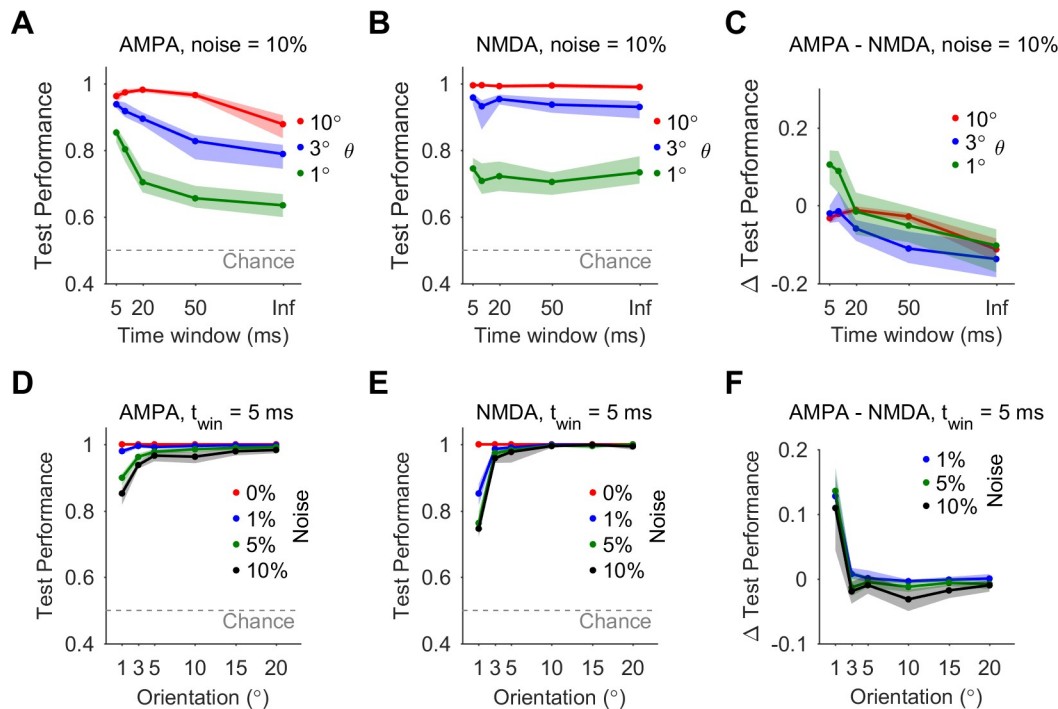

**Fig 3. Integration of fast-decaying (AMPA) synapses over short time window robustly discriminates fine orientations.**
**A.** Discrimination test performance of classifiers integrating fast-decaying (AMPA) synaptic inputs under noise = 10%, as a function of stimulus presentation time window. **B.** Same as A but for classifiers integrating slow-decaying (NMDA) synaptic inputs. **C.** The difference in test performance between classifiers integrating AMPA synaptic inputs and classifiers integrating NMDA synaptic inputs, for different time windows and noise = 10%. **D.** Discrimination test performance for classifiers integrating AMPA synaptic inputs over short integration time window (5 ms), for noise level = 0–10%. **E.** Same as D, but for classifiers integrating NMDA synaptic inputs. **F.** The difference in test performance between classifiers integrating AMPA synaptic inputs and classifiers integrating NMDA synaptic inputs, for integration time window of 5 ms and noise level 1, 5, 10%.

which exhibited areas of positive and null sensitivity that alternated between oppositely-tuned units (Figs 4C and 5C). In addition, the receptive field map showed that the synaptic integration relied on coincidence of activation from distal parts of the edge stimulus, where the difference between opposite-oriented edges is largest.

Whereas the number of key contributing neurons and the anti-correlation between oppositely-tuned units in classifiers that used slow-decaying (NMDA) synapses (Fig 5A and 5B, and see above) were similar to classifiers using AMPA synapses, the receptive fields of the classifier units were markedly different (Fig 5C). Classifier units that used AMPA synapses had distinct and narrow areas of positive or null sensitivity in their receptive field map (Fig 4C left), however classifier units that used NMDA synapses had broad areas of positive or null sensitivity in their receptive field map (Fig 5C left). This agrees with the utility of short-decaying (AMPA) synapses and long-decaying (NMDA) synapses for classification using coincidence detection or summation, respectively.

Edge-orientation classification relied on synaptic integration over multiple first-order neurons rather than simply on the tuning of single first-order neurons (Fig 6). The key first-order neurons were not always tuned to the edge-orientation in terms of peak rate (Fig 6B, first three example neurons), or in terms of response duration (Fig 6B, first and fourth example neurons). The key first-order neurons contributing to the edge-orientation classifiers had largely different receptive fields in size and mechanoreceptors innervation pattern, and stemmed from

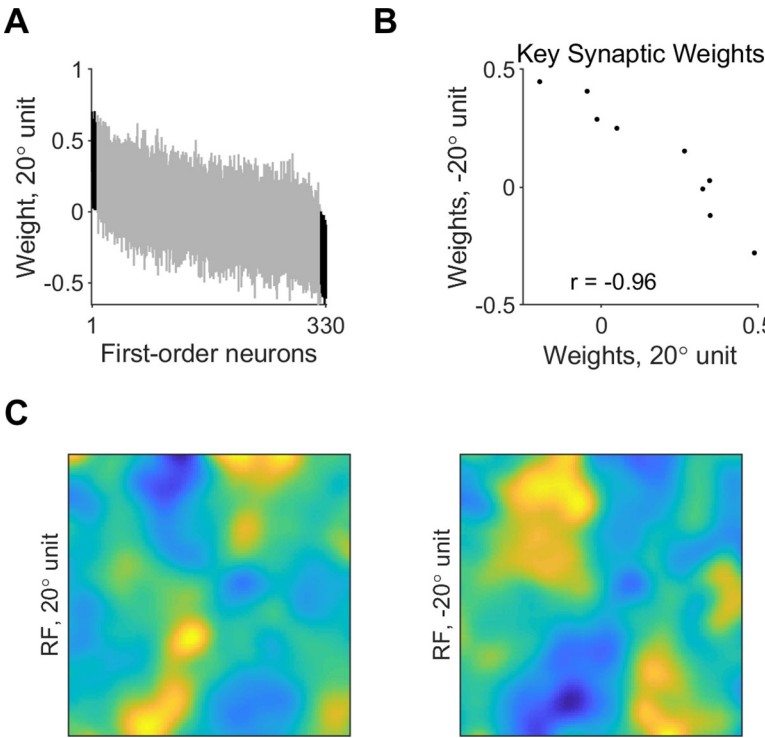

**Fig 4. Synaptic weights and receptive field of edge-orientation classifiers integrating fast-decaying (AMPA) inputs. A.** Synaptic weights from model first-order neurons onto classifier units using fast-decaying (AMPA) synapses and tuned to 20˚, with a presentation time window of 50 ms and noise level 5%. Shown are 95% confidence intervals of weights over 20 classifiers. Synaptic weights are shown in decreasing order of average weight over the 20 classifiers. Black–key synaptic weights, which were significantly different than 0. **B.** Key excitatory synaptic weights in example classifier units using AMPA synapses and tuned to 20 vs -20˚. **C.** Spatial layout of the synaptic weights in classifier units using AMPA synapses and tuned to -20 or 20˚, with a presentation time window of 50 ms and noise level 5%. Receptive fields represent averages over 20 classifiers.

different neurons from the pool of 15 core model neurons (Fig 6A). We also note that rotated variations stemming from the same core model neuron had largely different response firing rate curves (Fig 6B, fourth and fifth example neurons), attesting to the diversity of the neuronal population.

We inspected whether having a complex receptive field for first-order neurons contributed significantly to the accuracy of the edge-orientation classification. We utilized an alternative model in which the first-order neurons in the population innervating the patch of skin had a simple receptive field consisting of a single mechanoreceptor with responsivity radius as the average size of recorded receptive field (Fig 7A, see Methods). The simple receptive field led to a significant decrease in the edge-orientation discrimination accuracy of fine angles (1˚) in classifiers using short-decaying (AMPA) synapses regardless of the stimulus presentation time window (Fig 7B and 7D). For the long presentation time window, there was a significant decrease in accuracy also for coarse angles (10–20˚, Fig 7B). The accuracy of classifiers that used slow-decaying (NMDA) synapses did not change significantly with using a simple receptive field (Fig 7C). Thus, employing a simple receptive field resulted in an overall decreased edge-orientation discrimination acuity, and a decreased robustness of synaptic integration using AMPA receptors.

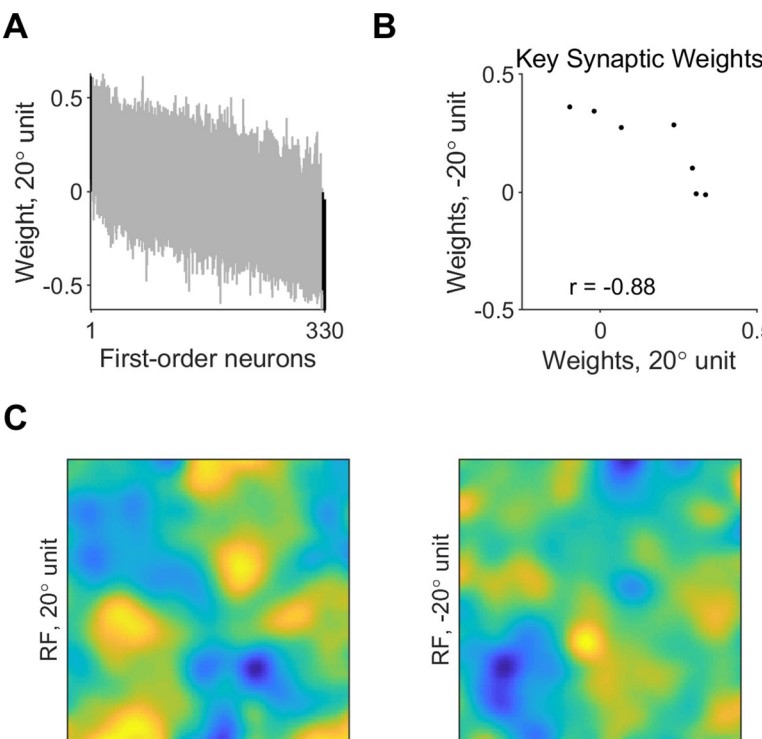

**Fig 5. Synaptic weights and receptive field of slow-decaying (NMDA) edge-orientation classifiers. A.** Synaptic weights from model first-order neurons onto classifier units using slow-decaying (NMDA) synapses and tuned to 20˚, with presentation time window of 50 ms and noise level 5%. Shown are 95% confidence intervals of weights over 20 classifiers. Synaptic weights are shown in decreasing order of average weight over the 20 classifiers. Black–key synaptic weights, which were significantly different than 0. **B.** Key excitatory synaptic weights in example classifier units using NMDA synapses and tuned to 20 vs -20˚. **C.** Spatial layout of the synaptic weights in classifier units using NMDA synapses and tuned to -20 or 20˚, with presentation time window of 50 ms and noise level 5%. Receptive fields represent averages over 20 classifiers.

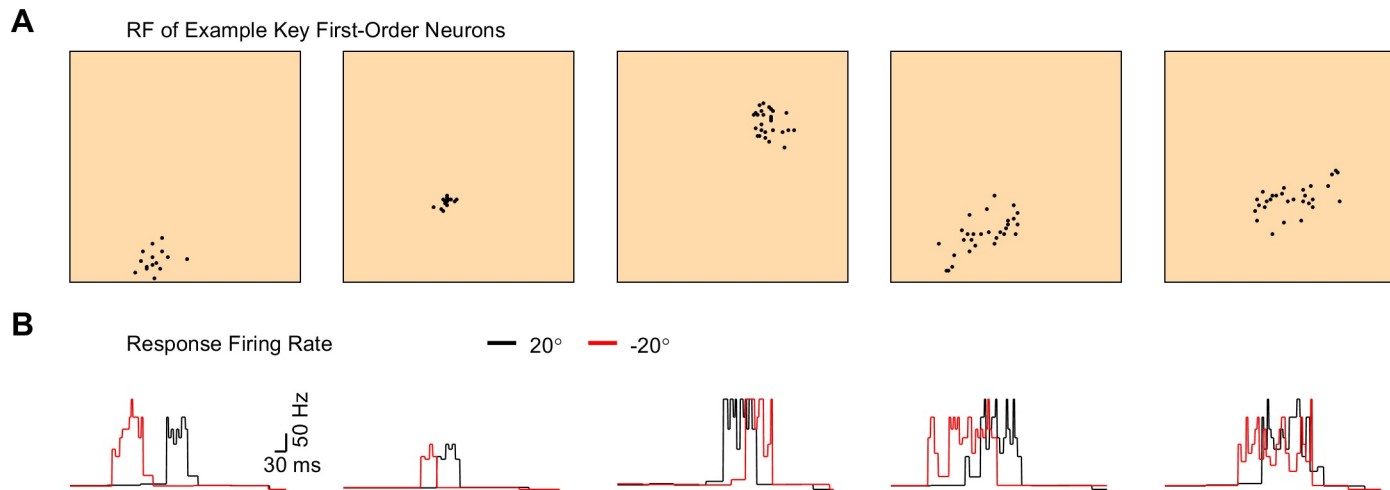

**Fig 6. Receptive field and response rate of first-order neurons contributing to edge-orientation classifiers. A.** The locations of innervated mechanoreceptors (black) in five example first-order model neurons from the population, which formed key excitatory synaptic weights in an edge-orientation classifier unit tuned to 20˚ and using fast-decaying (AMPA) synapses. The classifier was of the type shown in Fig 4. The skin patch dimensions are as in Fig 2. **B.** The firing rate of the example model neurons shown in A, in response to moving edges oriented 20˚ (black) and -20˚ (red), with a noise level of 5%.

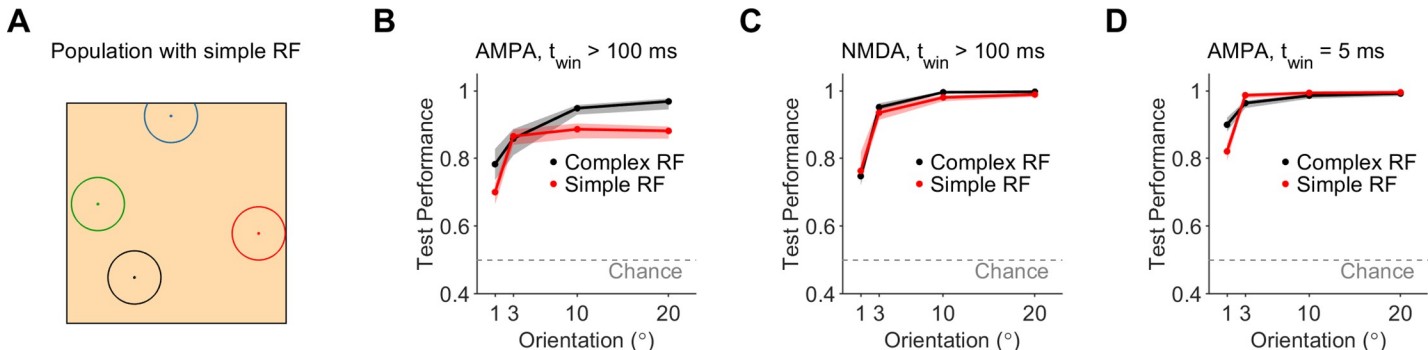

**Fig 7. Performance of edge-orientation classifiers that integrated over neurons with a simple receptive field. A.** Example receptive fields of four neurons from the population of first-order neurons, each of which innervated a single mechanoreceptor (see Methods). The patch of skin had dimensions as in Fig 2. **B.** Comparison of the test performance of classifiers that performed edge-orientation discrimination using synaptic integration over first-order neurons with a complex receptive field (multiple mechanoreceptors, black) or a simple receptive field (single mechanoreceptor, red). Classifiers used fast-decaying (AMPA) synapses, and were given a long (> 100 ms) stimulus presentation time window and a noise level of 5%. **C.** Same as B, but for classifiers using slow-decaying (NMDA) synapses. **D.** Same as B, but for a short (5 ms) stimulus presentation time window.

## Discussion

Our study provides new insight into tactile processing at the level of the first-order neuronal population. We derived spiking models of human FA-1 neurons innervating mechanoreceptors in the fingertips that were able to fit and predict responses to oriented edge stimuli with high accuracy. We then simulated the first-order neuronal population activity from the fingertip and showed that its readout via synaptic integration could underlie human ability of using edge-orientation information with high acuity and speed, as in the context of hand control. Although the primary focus of this study utilized edge-orientation as the choice stimulus to test discrimination accuracy, our simulation and models can be readily used to study the encoding of other stimuli such as different geometric shapes or textures, and different stimulus strengths/indentations.

We investigated edge-orientation discrimination using synaptic integration as the readout of model first-order neuronal population activity, and thus were able to provide additional mechanistic insight and testable predictions unavailable to previous models that computed using abstract features of the population response, such as mean firing rate, spike count, or spike latency [8–11,13]. Our modeling approach predicts that the high accuracy and speed of edge-orientation discrimination could be largely mediated by synaptic integration of first-order tactile neuronal population activity, as early as the level of second-order tactile neurons at the spinal cord or cuneate nucleus [16]. Moreover, our modeling approach differentiated between integration mediated by fast-decaying (AMPA) synaptic inputs (leaning more towards coincidence detection), and slow-decaying (NMDA) synaptic inputs (leaning more towards summation).

Integration of AMPA-like synaptic inputs, fast-decaying over a few milliseconds, enabled robust discrimination of fine angles within a short time window. Therefore, AMPA inputs may mediate initial rapid computations that could contribute to the fast and accurate responses that occur in the context of hand control [15]. The fast kinetics of AMPA inputs decreased their robustness in sustaining discrimination over longer time windows. Since the signal-to-noise ratio at the classifier's input was constant for a given noise level regardless of the presentation time window, the challenge for the classifier in discriminating edge-orientations using AMPA synaptic integration over longer time windows was more due to the

difficulty in finding the inputs that robustly coincided for the given edge orientation and did not for the other orientation.

Integration of NMDA-like synaptic inputs, slow-decaying over tens of milliseconds, maintained the robustness of discrimination when integrating inputs over longer time windows, and also supported discrimination for non-fine angles over short time windows. Therefore, NMDA inputs may mediate the discrimination of objects based on their macro-geometric features–for example, discriminating between your house key and car key when searching in your pocket. The differentiation between AMPA and NMDA input streams could be established for example via NMDA spikes triggered by particular input combinations in compartmentalized synaptic locations on the postsynaptic neuron dendrites [17]. In turn, the resulting outputs can be combined into a single spike train by multiplexed neural coding, whereby the two input streams yield spike trains with different frequency phase-locking or levels of synchronicity and can thus be multiplexed [18]. We note that although many of the stimulus presentation time windows were shorter than the NMDA decay time, and thus robustness over these time windows would be expected, our analysis involved longer presentation windows (>100 ms, often ~500ms for a full sweep of an edge over the patch of skin).

Our modeling effort and the underlying empirical data are based on tactile stimuli passively applied to the skin. Although this approach lets us precisely control the stimulus and thus understand the inputs to the nervous system it does not take into account that, under normal circumstances, a person would actively control how they move their hand over a surface or object when trying to extract information about it [19]. In the context of the present work it is important to distinguish between two modes of closed-loop control involving tactile sensation. Synaptic integration of AMPA inputs could serve for the fast and accurate ($< 3°$) extraction of fine features during object manipulation [15] that are critical to the demands of online hand and digit control [2,6,15,20]. In contrast, integration of NMDA inputs can serve in the haptic exploration of objects in terms of extracting macro-scale features [21,22] or establishing a robust perception of spatial features, a process which evolves relatively slowly and for which human sensitivity is ~10–25° [23–25], and where there appears to be limited benefit of active exploration over passive sensing [23,26,27]. This is not to say that active touch cannot provide perceptual benefits, just that such benefits typically arise in the context of haptic exploration of objects as a whole rather than their fine spatial features [21,22,28].

Our models also suggest a tendency for sparseness and the involvement of a small set of key synaptic inputs from first-order neurons, supporting previous empirical work [29] and suggesting that imposing a sparseness constraint on the synaptic weights may improve discrimination performance [30]. The reliance on a small set of key inputs can enable utilizing inputs from the other neurons for discriminating different orientations or stimulus features, as is also demonstrated in our two-orientations discrimination task by the non-overlapping sets of key neurons that contributed excitatory weights to classifiers tuned to opposite orientations (Fig 4B). Previous studies also showed opposing areas of sensitivity in the receptive fields of cortical neurons [31]. Our models predict similar properties for second-order neurons that integrate first-order population activity, whereby key synaptic connections would be anticorrelated between oppositely-tuned neurons. This prediction about the receptive field structure of second-order neurons could be further refined in the future by examining the discrimination of more than two orientations at a time, or the discrimination of different types of stimuli such as geometric shapes.

Our single-neuron models required ~20 mechanoreceptors on average to fit the spike response, which agrees with previous empirical estimates of the number of Meissner corpuscles converging onto a single FA-1 neuron [4]. We expect that a more complete set of constraints on the model neuron in the future would further increase the match between model

and experimental estimates. Compared to previous simpler models that predicted the average response firing rate using convolution of the recorded receptive field responsivity [6], our model neurons enabled a good prediction accuracy of single-trial response spiking rate and included physiological detail of mechanoreceptor innervation and spiking. An example of the added value of the nonlinearity of spiking was that our models exhibited a variety of peak rates in response to different edge orientations, consistent with previous empirical results [6,7], thus indicating that the diverse response could be reproduced with a physiologically-realistic model neuron that used spiking and resetting [32–34], and did not depend on analogue summation used by previous simpler models of human tactile receptive field responsivity [6]. Another example of the added value of spiking compared to the previous simple models, was the ability to reproduce the sharp spatiotemporal features of the firing response rate curve. This sharpness endows the neuron with a high spatiotemporal diversity in its stimulus response, and thus a higher computational power. The distinct events of high frequency response are also expected to translate to a sharply differing synaptic summation despite the smoothing effect of the synapses. Overall, this richness arises because neuronal spike output is not only a function of the present input but also of the neuron's previous spikes–a demonstration of how spiking increases a neuron's computational power [35].

Alternative spiking schemes, such as summation or spike mixing [36], were not as successful in reproducing the experimental firing. Models that used spike mixing tended to fire at high rates throughout the stimulus response as they lacked the toning-down effect of spike reset. Models that used summation were unable to reproduce the sharp transitions in firing rate seen experimentally during stimulus response. It remains to be seen if a spike reset scheme applies to other types of first-order tactile neurons, or in response to stimuli of lower intensity, which could instead involve a "mixing" of the spike trains [36].

In this work we have studied synaptic integration over the FA-1 neuronal population. As SA-1 neurons exhibit a transient rapid firing similar to FA-1 neurons and thus are able to report edge features [6], we expect that edge-orientation discrimination using AMPA synapses from the SA-1 population will work mostly similar to what we have observed for FA-1 neurons. In contrast, the sustained firing of SA-1 neurons after the initial rapid firing may decrease discrimination accuracy using NMDA synaptic inputs due to their summation, but since the sustained firing rate is relatively low the effect may not be large.

Our models included skin mechanics only implicitly, and thus we kept the distance parameters of mechanoreceptor responsivity free in the model optimization. Although this simplification was sufficient for fitting and predicting spike responses in our data, our data-driven model optimization framework can be extended to include explicit components of skin dynamics as used in other tactile neuron models [10,34].

We have generated a neuronal population using rotations of model neurons from a set of 15 neurons. Using populations derived from smaller sets of neurons did not influence the results. In addition, as shown in Fig 6, rotated versions of a model neuron had different response firing patterns, peak firing rates, and response duration, thus attesting to the diversity of the model neuronal population.

We have implemented additive noise at the level of the stimulus. The response of FA-1 and second-order cuneate neurons shows little variability [29,37,38], therefore there seems to be no significant noise at the level of mechanoreceptor to first-order neuron, or the synaptic projections from first-order neurons. While stimulus noise in the experimental data was < 1%, the larger levels of noise that we have investigated could represent other sources of noise such as variability in finger positioning during tasks. These in turn can account for the behavioral noise measured in tactile discrimination tasks [15]. While our choice of noise implementation involved a reasonable size of skin patches that were modified by noise, which also

corresponded to the size of the smallest stimulus in the experimental dataset (the dot stimulus) [6], in this work we have focused on the effect of noise level. Future studies could investigate different types of noise such as noise in the first-order spike trains, or varying the size of the skin patches that are being modified by the noise.

As some previous studies utilized a simple receptive field to model first-order neurons [13,14], we examined how the simple receptive field may affect the discrimination accuracy. We show that integrating over neuronal populations with simple receptive fields decreased the edge-discrimination accuracy of fine orientations regardless of the type of synapse or the stimulus presentation time window. Furthermore, we delineated cases where the simple receptive field performed comparably to the complex receptive field (e.g. for discriminating coarser orientations using NMDA synaptic integration). While the above studies achieved a comparably high discrimination accuracy by inputting the population activity into machine learning encoders, our study examined the encoding accuracy afforded by synaptic integration, to better gauge the computational power afforded by physiological synaptic mechanisms rather than the theoretical decoding limits. It is of general interest to further examine why the human tactile system has evolved complex receptive fields and the precise functional advantages complex receptive fields afford, for example using tasks of increasing difficulty or enforcing the kind of temporal constraints that arise during real world hand control.

## Methods

### Electrophysiology data

We used microneurography data of spike recordings from first-order tactile neurons, which was previously published [6]. The data consisted of 19 fast-adapting type 1 (FA-1) neurons from 13 human subjects (similar proportions of male and female). We utilized 15 of the 19 neurons, omitting two neurons whose receptive field was larger than the inter-stimulus spacing, and two neurons that had a low stimulus/non-stimulus response ratio. The recorded neurons innervated the glabrous skin of the index, long or ring finger. Neurons were stimulated via a rotating drum embossed with dots and edges at different orientations. Each stimulus thus consisted of a single edge at a time, moving over the receptive field at a speed of 30 mm s$^{-1}$.

### First-order tactile neuron models

Model neurons innervated a subset of mechanoreceptors in a patch of skin modeled as a grid of mechanoreceptors located at 0.1 mm intervals [4]. The input from the model mechanoreceptor was proportional to the stimulus amplitude (indentation) and decreased with the distance from stimulus, following a sigmoidal function:

$$I_{MR} = A_{stim} w_{MR}(1 - 1/(1 + \exp(-5(d/r_1 - 1)))) \qquad d \leq r_1 + r_2; \qquad 0 \qquad d > r_1 + r_2 \quad (1)$$

$A_{stim}$ was the stimulus amplitude, or tactile edge indentation, which was 0.5 mm as in the experimental data. $w_{MR}$ was the input weight of the mechanoreceptor, which was the same for all mechanoreceptors and set as twice the maximal firing rate of the experimental neuron. Thus, a stimulus indented 0.5 mm moving across a mechanoreceptor provided sufficient input to support the maximal firing rate of the recorded neuron. $d$ was the distance (in mm) of the stimulus from the mechanoreceptor. $r_1$ was the sigmoidal half-height distance, and $r_1 + r_2$ was the extent of the mechanoreceptor responsivity.

The model neuron innervated each mechanoreceptor with a dedicated axonal branch [4] and spikes were initiated at each axon terminal (Fig 1A) [32–34,36,39,40]. Spike generation depended on mechanoreceptor input and the time from the last spike, following a linear

relationship between mechanoreceptor input and spike rate (with gain = 1), and saturation at the maximal rate of the recorded neuron. The model neuron output followed a "reset" scheme, whereby spikes from one spiking zone propagated retrogradely to the other spiking zones and reset their spike initiation [32–34]. The model also included spike adaptation, whereby each model neuron axonal branch fired only when input from mechanoreceptor was increasing, and was silent when the input remained the same or decreased. We used a spike threshold of 0.01 mm stimulus indentation [3]. We simulated the model neuron response to moving edge stimuli at Δt = 1 ms intervals. The model was implemented and simulated in Matlab (Mathworks).

## Neuron model fitting

We constrained the models with spike recordings of the response to edges of four different orientations (± 22.5, ± 45˚). The error measure for the goodness of fit was $R^2$ computed between the recorded and model spike rate curves:

$$R^2 = \langle (SR_m - SR_o)^2 \rangle / \langle (SR_o - \overline{SR_o})^2 \rangle \tag{2}$$

Where $SR_m$ is the spike rate time-series of the model neuron, and $SR_o$ is the spike rate time-series of the recorded neuron. The error was calculated for each edge orientation and averaged over the orientations to produce a ranking of the model.

We used a genetic algorithm [41–43] to search for the free model parameters, which were the locations of the mechanoreceptors innervated by the model neuron, as well as the two distance parameters of the mechanoreceptor responsivity ($r_1$ and $r_2$, see above). The set of possible mechanoreceptor locations was delineated by the area of responsivity of the recorded neuron to a small dot stimulus (Fig 1B, gray). The search limits for the mechanoreceptor response distance parameters were [0.05,1] mm for $r_1$ and [0.2,1] mm for $r_2$. During optimization, a population of 100 models was mutated with a probability of 0.1, and underwent crossover with a probability of 0.1 at each iteration. Models were rated based on how well they fitted the recorded response to the four edges (see above). We implemented the algorithm in Matlab, and optimization runtime of 500 iterations on 4 processors was 1 hour on average.

## Model neuron cross-validation and selection

For each recorded neuron, we ran separate model optimizations using different number of innervated mechanoreceptors: 10, 20, 30, or 40. We selected from the resultant models the one that best fitted the training data with fewest mechanoreceptors, and that had the best cross-validation accuracy for an edge-orientation of 0˚ (which was not used during fitting). We first selected the set of best models in terms of training data fitness, i.e. models with $R^2$ within 5% of the maximal fit for each of the four edge orientations used to train the models. From this set, we selected the models that had the highest cross-validation accuracy ($R^2$ within 5% of the maximal cross-validation accuracy) and fewest mechanoreceptors (within 5 mechanoreceptors from minimum across the models). From these models, we selected the model with the best cross-validation accuracy as the model for the recorded neuron.

## Model neuron testing

After the optimization, we tested models using the spike recordings in response to edges oriented 30˚. We compared the prediction to null models, in which the mechanoreceptor locations were shuffled across the area of responsivity (see above). The error measure was $R^2$, as

described above for the fitness. In addition, we compared the model prediction to that obtained by simply using the recorded response to the nearest edge (22.5˚).

## Model neuronal population

We generated model first-order neuronal population innervating a fingertip as randomly-rotated variations of the 15 model neurons. We tiled the model neurons over a 12x12 mm patch of skin, so that they innervated the patch at a uniform density of 140 neurons/cm$^2$ [2]. This resulted in 330 model neurons innervating mechanoreceptors in the skin patch.

## Edge-orientation discrimination

We simulated the activity of the neuronal population during a task of discriminating between two edges sweeping over the fingertip and oriented at -θ or θ, where θ = 1, 3, 5, 10, 15 or 20˚. To discriminate the edge-orientations, we used classifiers comprised of two units, one tuned to -θ and another tuned to θ. Each unit performed a synaptic readout of neuronal population activity, via a weighted sum of the postsynaptic potentials (PSPs). In accordance with the physiology of first-order neuron projections, we used synapses with time constants that corresponded to AMPA and NMDA synapses [44,45]. The PSP vector from each first-order neuron was computed by convolving the spike train with a PSP waveform, which was the difference of two exponentials:

$$\exp\left(-\frac{t}{\tau_{decay}}\right) - \exp\left(-\frac{t}{\tau_{rise}}\right) \tag{3}$$

Where $\tau_{rise}$ = 0.5 ms, and $\tau_{decay}$ was either 3 ms as in AMPA receptors [46] or 65 ms as in NMDA receptors [47]. We also allowed fast-decaying and slow-decaying inhibitory synapses in the classifiers, which would correspond to feed-forward inhibition via $GABA_A$ and $GABA_B$, respectively, with similar time constants to their excitatory counterparts [48,49]. Discrimination was based on the classifier unit with maximal PSP value over the time series, corresponding to a linear integration with threshold. We trained the model networks using a genetic algorithm to search for the input weights from the first-order neurons [50]. For the genetic algorithm, we used weights limits of [–1, 1], a population of 100 models, mutation probability of 0.1, crossover probability of 0.1, and 200 iterations.

We investigated the discrimination accuracy under different levels of additive stimulus noise: 0, 1, 5, or 10%. Noise was added as 0.4 x 0.4 mm patches that varied in amplitude. The size of the noise patches was chosen to match the size of the smallest stimulus type (dot) in the experimental data (see above), and also in terms of reasonable surface irregularities. For example, a noise level of 10% involved addition of noise patches of random amplitudes ranging between -10 and 10% of stimulus amplitude. While the additive noise could reduce stimulus indentation at some points, we forced the resulting stimulus to be strictly non-negative by setting the resulting stimulus amplitude in the 0.4 mm patch to 0 whenever the noise application reduced it below 0. For each noise level examined, we trained classifiers using training data that had that noise level, and tested it on data that had the same noise level. We simulated 100 trials for each orientation, where trials had the same noise level but differed in the randomized noise added to the stimulus (see above). We picked 50 trials of each orientation at random for training the model networks (100 trials in total), and used the remaining 50 trials of each orientation for testing the model networks. For each task, we generated 20 model networks using different random subsets of training/testing trials. The performance measure was success discrimination rate over trials.

## Stimulus presentation time window

We investigated a range of time windows available for the edge-orientation discrimination. In all cases, the edge passed across the receptive field at the same speed as in the experimental data (30 mm s$^{-1}$). In the unlimited window case, the edge passed over the skin patch fully from one end to the other. During shorter time windows, the edge passed for a limited time (5, 10, 20, or 50 ms) around the center of the skin patch. Thus, longer presentation time windows meant that the edge passed across larger parts of the skin patch, whereas shorter presentation time windows meant that the edge passed across smaller parts of the skin patch.

## Simple receptive field

We implemented alternative models of first-order neurons using a simple receptive field, innervating a single mechanoreceptor with responsivity function as in Eq 1 above, and with a responsivity radius $r_1$ = 0.05 mm and $r_2$ = 1.45 mm. The simple receptive field thus had a size similar to the average recorded neuron, and involved a symmetric responsivity profile unlike that of the recorded complex neuron.

## Statistical tests

We determined correlation using Pearson's correlation coefficient. We tested for statistical significance using either paired or two-sample t-test, in Matlab. P-values $< 0.05$ were deemed significant. When estimating 95% confidence intervals, we used bootstrap estimation of the mean value in Matlab, and corrected for multiple comparisons using Bonferroni method (multiplying the $p$ value by the number of comparisons).

The models, algorithms, simulation code and data used in this study are available online on ModelDB (https://senselab.med.yale.edu/modeldb/), accession number 266798.

## Author Contributions

**Conceptualization:** Etay Hay, J. Andrew Pruszynski.

**Formal analysis:** Etay Hay, J. Andrew Pruszynski.

**Funding acquisition:** Etay Hay, J. Andrew Pruszynski.

**Investigation:** Etay Hay.

**Methodology:** Etay Hay.

**Software:** Etay Hay.

**Supervision:** J. Andrew Pruszynski.

**Visualization:** Etay Hay.

**Writing – original draft:** Etay Hay, J. Andrew Pruszynski.

**Writing – review & editing:** Etay Hay, J. Andrew Pruszynski.

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
