## [Decision Letter · Decision Letter 0]

5 Oct 2019

Dear Dr Hay,

Thank you very much for submitting your manuscript 'Orientation processing by synaptic integration across first-order tactile neurons' for review by PLOS Computational Biology. Your manuscript has been fully evaluated by the PLOS Computational Biology editorial team and in this case also by independent peer reviewers. The reviewers appreciated the attention to an important problem, but raised some substantial concerns about the manuscript as it currently stands. Please attend to all of the reviewers comments. Note especially Reviewer 3's last major comment. It is important to provide some discussion in order to compare with a previous model of tactile processing with very different properties. While your manuscript cannot be accepted in its present form, we are willing to consider a revised version in which the issues raised by the reviewers have been adequately addressed. We cannot, of course, promise publication at that time. 

Sincerely,

Blake A. Richards

Associate Editor

PLOS Computational Biology

Lyle Graham

Deputy Editor

PLOS Computational Biology

[LINK]

Reviewer's Responses to Questions

**Comments to the Authors:**

Reviewer #1: The paper describes the development and optimization of a model of first order spiking sensory neuron responding to tactile stimuli of varying orientations. Optimization is carried out by varying mechanoreceptor location and their receptive fields to fit human neuronal recordings. Ultimately, the fitted model is claimed to be predictive for unforeseen stimuli.

Having the model for spiking patterns devised, the authors created a classifier that relies on population of modeled neurons to determine the stimulus orientation. In this context, the difference between slowly integrating NMDA based classifier and a fast AMPA based one is demonstrated with AMPA synapses outperforming NMDA with high resolution orientation detection.

Major comments:

1. The model was trained on non-trivial positive/negative angles and then tested only on a zero angle. It is recommended that the authors will test the model against several angles and compare and analyze the dependency of model performance on stimulus angle. Furthermore, from the methods if sounds like the test set (angle zero) was used in the final model picking. This should be clarified to ensure appropriate separation between fitting and cross validation. Particularly alarming is the statement at the bottom of page 18 saying that the model was selected based, among other things, on its test performance.

2. Another point, which claimed to be a major advance (in intro/discussion), is relying on the actual spike trains for modeling. It should be clarified what part of the modeling process will fail if a model based on rates or on first neuron to spike or other less detailed description of neuronal activity is considered. In particular, given an integrator that the authors use at the input to the classifier essentially performs a translation from spikes to rate..

3. The authors claim that the AMPA based model works like a coincidence detector, whereas the NMDA based model “may mediate the discrimination of objects based on their macro-geometric features". Beyond Fig3A-C, the comparative results are based on accuracy in the presence of additive noise, which the authors acknowledge to be a minor issue for tactile perception. To make the case of AMPA vs. NMDA strong, and given the central role noise plays in the comparison, it is recommended to:

- explain what noise level was used for training. If training was done with zero noise, please redo it with the noise level at which Fig 3 is presented (i.e. 10%).

- describe the noise protocol in more details (in Methods), including noise spectrum and the way the non-negativity is enforced.

- repeat FIG4 for the NMDA receptive fields, and further repeat it for fine orientations (+/-1 or +/-3deg) to emphasize the synaptic mechanism of coincidence detection.

- analyze further the deterioration of AMPA results for long integration time (in Fig. 3A). - quantify the Signal-to-Noise ratio at the classifier's input as a function of the integration time

4. The model is an open-loop model. In typical tactile sensing, humans control their scanning kinematics, probably as a function of the tactile input (a closed-loop control). The authors should refer to the difference between these two modes of perception and discuss how their results can contribute to the understanding of closed-loop tactile perception.

Minor points:

1. In the discussion it is suggested that AMPA receptors allow fast discrimination while NMDA receptors are responsible for more fine tuning-but the results suggest that that AMPA rectors are superior to NMDA at short time scales for small angles only (fig.2E.fig.3C,F).

2. The change in performance with time is related to AMPA and NMDA decay constants-how does these relate to the frequency of stimulus presentation (and possible adaptation affects).

3. Comparison to ref. 6 (page 15): what figure of merit is compared?

4. Argument for sparseness in figure 4A: Does that mean that the majority of 330 neurons are redundant? Or are they used for fine orientations (e.g. +/-1deg.).

Reviewer #2: In this manuscript, Etay and Pruszynski built spiking models of first-order tactile neurons and explored if/how a decoder extracts orientation information from a population of such neurons. The first-order neurons have complex receptive fields that endow them with orientation tuning. I think the results are interesting and, from what I can tell, the modeling has been conducted carefully and thoroughly. I do, however, have some concerns and suggestions for how the improve the paper.

Main concerns:

-The Introduction explains that both FA-1 and SA-1 neurons play a role in sensing fine spatial details. But for the rest of the paper, only FA-1 neurons are considered. This leaves me wondering how the FA-1 and SA-1 neurons might differ, and specifically whether they contribute differentially to orientation tuning (or would be decoded differently). Especially when considering the timescales of AMPA and NMDA synapses, I’m wondering if the timescale of SA-1 responses (if different from FA-1 responses) would cause decoding to be better/worse for AMPA/NMDA synapses. At the very least, there should be some discussion of these issues.

- I’m unclear whether the reset scheme used to model spike generation across different axon branches is critical. Is that the only scheme that works to reproduce the experimental spike trains, or do alternative schemes (e.g. summation) also work? If both work, are there any implications for subsequent decoding?

-The authors developed 15 different neurons for primary afferent neurons. A patch of skin is modeled using 330 models, each of which is randomly rotated. This means that each model is used 22 (or more) times. Notwithstanding the rotation, I worry that re-using identical models might introduce unintended structure that would not occur naturally and which might affect decoding. I suggest morphing the existing models so that they have the same RF structure in a statistical sense, but are not identical, and conducting some test simulations to rule out key differences from the random rotation approach. Alternatively, would building a 330-cell network with just one sample neuron (instead of 15, but still rotating them), yield equivalent results to the current approach. If the test simulation don’t raise concerns, I do NOT see any need to repeat all the simulations.

-The issue of a stimulation time window raised questions for me. The RF is mapped experimentally by sweeping a single edge (or grating?) across the RF, correct? If so, does the “time window” relate to the sweep speed? There is room to clarify how the stimulus is implemented in the model and what certain parameters mean in a physiological sense.

-Figure 4 explains that only a small number of sensory neurons (8 to 11, out of 330) are used to decode orientation as inferred from synaptic weights. It is mentioned that the weights of key synapses are anticorrelated between classifiers tuned to opposite orientations. This leads me to suspect that all the sensory neurons providing input to each classifier (i.e. synapses with significant weight) all have the same orientation tuning, and the classifier simply inherits that tuning. In other words, optimizing the synaptic weight amounts to finding the correctly tuned sensory neurons, connecting to them, and not connecting to sensory neurons with other tuning. Is that correct? If so, the result seems a little trivial. More sophisticated downstream circuitry (e.g. with inhibition driven by sensory neurons with orthogonal orientation tuning) might refine the decoding, but this isn’t considered. Please comment.

Other points:

-Articles like “the” or “a” are often missing. Examples (p 3): with AN axon spike initiation zone; we used A genetic algorithm; best fit of THE model neuron. There is room to improve the writing in this regard.

-I do not see any supplementary files providing the data. More importantly for this sort of paper, I do not see any statement about the availability of the code used for modeling.

Reviewer #3: The receptive fields of first order tactile neurons have multiple hotspots. In a previous paper, the senior author has proposed that this receptive field structure contributes to the extraction of the geometric features of a stimulus, namely edge orientation. This new study first develops a simple model of the response of individual first order neurons. The main innovation in the model is that it incorporates multiple mechanoreceptors, whose spiking response is then integrated using a winner-take-all mechanism. This model can faithfully reproduce the responses of first order neurons to scanned edges. Second, the response of a population of first order neurons to scanned edges is simulated using the model. The output of the neurons is then convolved with one of two filters – one designed to mimic fast decaying AMPA-mediated currents, the other designed to mimic slow decaying NMDA-mediated currents. The output of the resulting population of simulated second order neurons is then used to classify the orientation of the stimulus. Classification performance is found to depend on stimulus duration and noise and also depends on which filter is used to generate the second order signals.

How first order neurons integrate signals from different mechanoreceptors is an interesting and important question, which this study purports to address. However, this aspect of the study is then combined with a different one, which is to assess whether orientation can be decoded from these simulated signals. Neither model – encoding or decoding – is validated. This work rather thus constitutes a proof of principle rather than a demonstration.

Major comments

1. The model is fit to scanned edges and tested on scanned edges. It is difficult to evaluate the performance of the model and compare it to previous models. For example, previous models of first order neurons achieved comparable performance in predicting neural responses to a wide variety of stimuli, including edges. How do the models compare in terms of performance? This is important to establish the importance of the multi-receptor models (the main innovation in the encoding model) compared to single-receptor models.

2. One of the questions about coding in first order neurons is the extent to which these responses convey stimulus information in a reliable way. In this study, all the edges are scanned at a single speed. Robustness to noise is tested, with Gaussian noise. The dependence of the (simulated) neural responses to noise are poorly characterized. What happens when you change the size of the noise patch, e.g.? Only the consequence of the noise on decoding performance is discussed. The noise seems to be largely arbitrary (Why .4 x .4 mm patches? for example). More importantly, to what extent would the models be able to accommodate edges with different geometries, scanned at different speeds (on which the senior author has previously published), or indented into the skin? How to first order neurons respond to these stimuli? Can the orientation of these stimuli be decoded using a single decoder?

3. How does orientation decoding performance compare to previously published performance in a comparable exercise with fundamentally different encoding and decoding models (Delhaye et al., JNP, 2018)?

Minor comments

Does it make sense to run an analysis of performance at different time windows with “NMDA” currents given their long integration times?

**Have all data underlying the figures and results presented in the manuscript been provided?**

Reviewer #1: Yes

Reviewer #2: No: Perhaps I overlooked something, but I do not see any supplementary files providing the data. More importantly for this sort of paper, I do not see any statement about the availability of the code used for modeling.

Reviewer #3: Yes

PLOS authors have the option to publish the peer review history of their article (what does this mean?). If published, this will include your full peer review and any attached files.

Reviewer #1: No

Reviewer #2: No

Reviewer #3: Yes: Sliman Bensmaia

---

## [Decision Letter · Decision Letter 1]

20 Aug 2020

Dear Dr Hay,

Thank you very much for submitting your manuscript "Orientation processing by synaptic integration across first-order tactile neurons" for consideration at PLOS Computational Biology. As with all papers reviewed by the journal, your manuscript was reviewed by members of the editorial board and by several independent reviewers. The reviewers appreciated the attention to an important topic. Based on the reviews, we are likely to accept this manuscript for publication, providing that you modify the manuscript according to the review recommendations. The reviewers have a few remaining minor items, all of which we believe you can attend to.

Sincerely,

Blake A. Richards

Associate Editor

PLOS Computational Biology

Lyle Graham

Deputy Editor

PLOS Computational Biology

[LINK]

Reviewer's Responses to Questions

**Comments to the Authors:**

Reviewer #2: I am happy with the revisions. My initial set of concerns have been adequately addressed in the revised manuscript and in the response to reviewers. My remaining points are all quite minor.

-The author summary seems like recycled version of Abstract. It would be helpful to use the Author Summary to describe the main findings and their impact, rather than re-hashing the approach, as already outlined in the Abstract.

-Second-to-last paragraph of Introduction: the authors seem to imply that central neurons are upstream of peripheral neurons. I always consider the postsynaptic neuron to be downstream. If there is any room for confusion here, I would suggest replacing “downstream” with “central” or “second-order”.

-Figure 1 legend: I think it is clearer to explain that each mechanoreceptor receives its own “axonal branch”, as opposed to its own “axon”, as those various branches come together to form a single axon projecting to the CNS. This terminology comes up in a couple other places, and though minor, the suggested change might help avoid confusing certain readers.

-Above Figure 3 legend: “…increased the robustness of discrimination of classifiers [that] used fast-decaying…”

-Figure 2 legend: “…each of which innervat[ed] a single mechanoreceptor”

-First paragraph of Discussion; “used to study the encoding [of] other stimuli”

-Fourth paragraph of Discussion: the authors suggest that “differentiation between AMPA and NMDA input streams could be established for example via compartmentalized synaptic locations on the postsynaptic neuron dendrites”. It is not clear to me how that would happen if both types of synaptic inputs evoke spikes that are interspersed in the same neuron. I appreciate how the authors might decode using the PSP amplitude in different dendritic branches, but that is ultimately not a biologically realistic decoding scheme. Could the authors please clarify this idea.

Reviewer #3: I'm satisfied with the authors revisions. One question: Can we glean, from this study, a prediction about the RF structure of cuneate neurons? Is that what is shown in Figures 4 and 5C?

**Have all data underlying the figures and results presented in the manuscript been provided?**

Reviewer #2: **No: **The authors state that the model code will be made available. I think that is what's critical for this study.

Reviewer #3: Yes

PLOS authors have the option to publish the peer review history of their article (what does this mean?). If published, this will include your full peer review and any attached files.

Reviewer #2: No

Reviewer #3: **Yes: **Sliman J Bensmaia
---

## [Editor Report · Decision Letter 2]

3 Sep 2020

Dear Dr Hay,

We are pleased to inform you that your manuscript 'Orientation processing by synaptic integration across first-order tactile neurons' has been provisionally accepted for publication in PLOS Computational Biology.

Best regards,

Blake A. Richards

Associate Editor

PLOS Computational Biology

Lyle Graham

Deputy Editor

PLOS Computational Biology

---

## [Editor Report · Acceptance letter]

28 Oct 2020

PCOMPBIOL-D-19-01342R2 

Orientation processing by synaptic integration across first-order tactile neurons

Dear Dr Hay,

I am pleased to inform you that your manuscript has been formally accepted for publication in PLOS Computational Biology. Your manuscript is now with our production department and you will be notified of the publication date in due course.

With kind regards,

Matt Lyles
